# Influence of Tillage Systems and Cereals–Legume Mixture on Fodder Yield, Quality and Net Returns under Rainfed Conditions

**Sunyad Sohail [1], Muhammad Ansar [1], Milan Skalicky [2], Allah Wasaya [3,*], Walid Soufan [4],**
**Tauqeer Ahmad Yasir [3], Ahmed M. El-Shehawi [5], Marian Brestic [2,6], Mohammad Sohidul Islam [7],**
**Muhammad Ali Raza [8] and Ayman EL Sabagh [9,*]**

1 Department of Agronomy, PMAS-Arid Agriculture University, Rawalpindi 46000, Pakistan;
   sunyad.sohail@gmail.com (S.S.); muhammad.ansar@uaar.edu.pk (M.A.)
2 Department of Botany and Plant Physiology, Faculty of Agrobiology, Food and Natural Resources,
   Czech University of Life Sciences Prague, Kamycka 129, 165-00 Prague, Czech Republic;
   skalicky@af.czu.cz (M.S.); marian.brestic@uniag.sk (M.B.)
3 College of Agriculture, Bahauddin Zakariya University, Bahadur Sub-Campus Layyah,
   Layyah 31200, Pakistan; tayasir@yahoo.com
4 Plant Production Department, College of Food and Agriculture Sciences, King Saud University, P.O. Box 2460,
   Riyadh 11451, Saudi Arabia; wsoufan@ksu.edu.sa
5 Department of Biotechnology, College of Science, Taif University, P.O. Box 11099, Taif 21944, Saudi Arabia;
   a.elshehawi@tu.edu.sa
6 Department of Plant Physiology, Slovak University of Agriculture, Nitra, Tr. A. Hlinku 2,
   949-01 Nitra, Slovakia
7 Department of Agronomy, Hajee Mohammad Danesh Science and Technology University,
   Dinajpur 5200, Bangladesh; shahid_sohana@yahoo.com
8 College of Agronomy, Sichuan Agricultural University, Chengdu 611130, China; Razaali0784@yahoo.com
9 Department of Agronomy, Faculty of Agriculture, Kafrelehikh University, Kafr Elsheikh 33511, Egypt
* Correspondence: wasayauaf@gmail.com (A.W.); aymanelsabagh@gmail.com (A.E.S.)

**Abstract:** Livestock development in rainfed areas is slower due to the inadequate supply of nutritious fodder. Mono-cropping systems also have a negative impact on forage yield and nutrition as cereals are deficient in protein. Hence, there is a dire need to grow cereals with legumes to improve forage yield and quality. Therefore, a two-year field study was undertaken to evaluate winter cereal–legume forage and their mixtures viz. oats (cv. $PD_2$-$LV_{65}$), barley (Jau-86) and one legume viz. vetch (cv. Languedock) under different tillage systems viz. conventional tillage (moldboard plow+4-cultivation with tines) and conservation tillage (3-cultivation with tines). Crops were grown in pure stands as well as in mixtures with a 70:30 seeding ratio. The results revealed that the conventional tillage system performed better in terms of numbers of tillers/branches, leaf-to-stem ratio and green fodder yield than the conservation tillage system. However, the conventional and conservation tillage systems did not show a significant difference in terms of crude protein, acid detergent fiber and neutral detergent fiber. In the pure stands and cereal–legume mixtures, the oat–vetch mixture performed better in terms of plant height, leaf-to-stem ratio and green fodder yield. The maximum crude protein content was observed in the oat–vetch mixture, while the maximum acid detergent fiber and neutral detergent fiber were observed in the pure oat stands. In competitive indices, the land-equivalent ratio and competitive ratio showed the advantage of intercropping. In actual yield loss, results showed the positive value of barley and oats in mixtures, which reflects the advantage of intercropping in the rainfed areas. The economic analysis showed a greater net benefit from the conventional tillage than the conservation tillage system under rainfed conditions. On the basis of this investigation, an oat–vetch mixture and the conventional tillage system are recommended for higher tonnage of nutritious fodder in rainfed areas.

**Keywords:** tillage systems; cereal–legume mixture; fodder yield; acid detergent fiber; land-equivalent ratio

## 1. Introduction

Livestock production is an important part of agriculture, especially in rainfed areas. Raising field crops for grain purposes, however, can be risky because of a lack of water required for crop growth and development. Rural communities meet their dietary requirements for meat, milk, etc. from livestock and obtain economic value from byproducts such as leather and wool. However, livestock may develop more slowly in rainfed areas due to a shortage of nutritious fodder [1].

Inceptisols and aridisols are common soil orders in arid regions with silty loam to sandy loam texture, as well as uneven and sloping terrain [2]. Soil erosion, land degradation and surface crusting are common problems in such areas [3]. Another major challenge is soil moisture conservation [4] because the evaporation rate exceeds precipitation. Farmers in this area use the conventional moldboard plow followed by 4–8 runs with the tine cultivator for in situ moisture conservation. Due to this continuous intensive plowing at the same depth, a hardpan often forms beneath the soil surface. This hardpan restricts the movement of the soil water and plant nutrients in deeper profile layers, and such intensive tillage systems encourage soil erosion [5,6]. On the other hand, conservation tillage, i.e., zero tillage, minimum tillage, direct drilling, as an alternative to intensive tillage involves minimal disruption to the soil while leaving at least 30% of the crop residues on the soil surface [7]. There are several benefits of conservation tillage for soil health, water conservation, crop production and accentuated environment [8].

Green fodder scarcity during the winter months is attributed to the slower growth of forage crops at this time. Farmers usually feed dry stalks of cereals to their animals, which are nutritionally poor [9]. Cereals and legumes are important forage crops because of their nutritional value, especially the protein in legumes and fiber in cereals [10]. However, fodder made from cereals alone is of low forage quality as it contains too little protein. Thus, it is necessary to prepare forage by combining a cereal crop with a legume crop to increase the protein content of the feed. This mixed cropping of certain annual legumes with cereals increases forage yield and quality [11]. Another problem in raising good quality forage is the fertility status of the soils, which in arid regions is low. The use of inorganic nitrogenous fertilizers is limited because of the uncertainty of rainfall and because chemical fertilizers are too expensive for farmers in the rainfed areas. They are using exhaustive crops like wheat and brassica as green fodder in winter and maize/sorghum or millet in summer. Hence, there is a dire need for intercropping legumes with cereals to produce high quality forage so that raising livestock can become a profitable, sustainable business in rainfed regions [12].

Given the importance and problems of tillage and cropping in rainfed areas, the current study was conducted to evaluate the forage crops, oats, barley and vetch, and their combinations oats + vetch and barley + vetch under different tillage methods to identify the most economical system for improving forage yield and quality under rainfed conditions.

## 2. Materials and Methods

### 2.1. Experimental Site

A two-year study (2013–2014 and 2014–2015) was conducted in the field at the university research farm (33° N and 42.72° E), Pir Mehr Ali Shah Arid Agriculture University, Rawalpindi, Pakistan during Rabi season. The region has a semi-arid climate, and two-year climatic data for the whole crop season are presented in Figure 1. The soil of the area is loamy and at 0–15, and 15–30 cm depth, respectively, it contains 0.62 and 0.52% organic matter, 1.42 and 1.09 ppm N (as $NO_3$), 6.67 and 6.23 mg kg$^{-1}$ available P, and 138 and 135 mg kg$^{-1}$ available K.

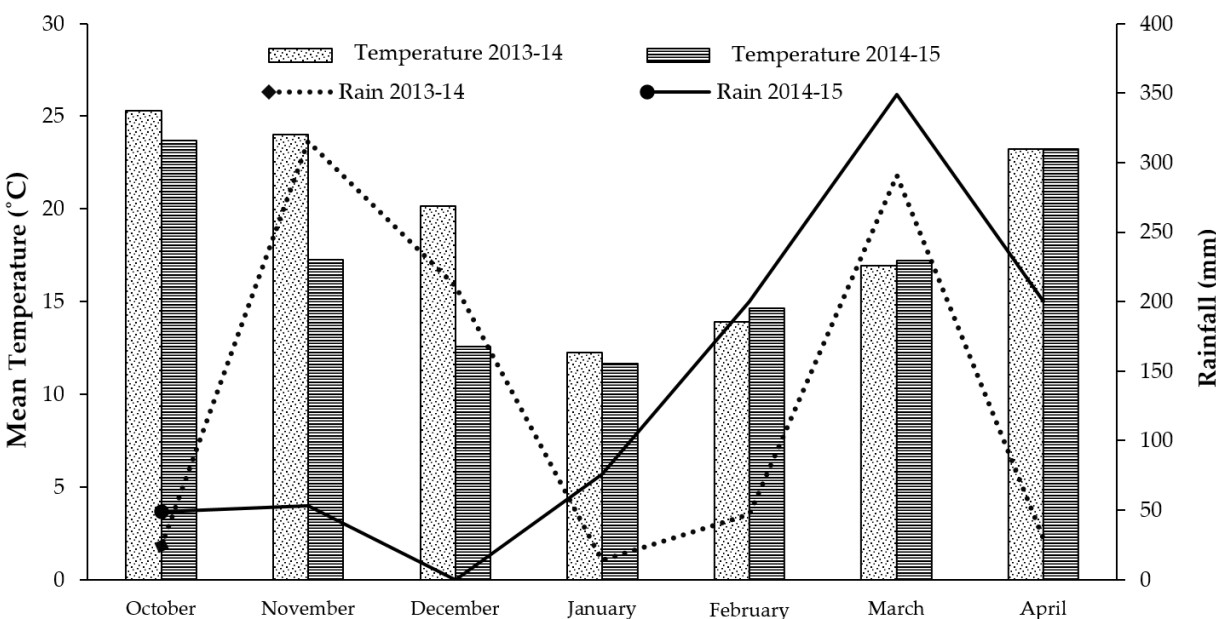

**Figure 1.** Rainfall (mm) and mean temperature (°C) 2013–14 and 2014–15.

### 2.2. Experimental Methods

Cereal–legume fodder was cultivated under two tillage systems: conventional, using a moldboard plow (MBP) and four passes with a tine cultivator and conservation, without plowing but with three passes with a tine cultivator. Two cereals, oat (cv. $PD_2$-$LV_{65}$) and barley (Jau-86), and one legume, vetch (cv. Languedoc), were grown in both pure and mixed stands. The experimental plot was laid out according to a randomized complete block design (RCBD) with strip plot arrangement and four replications. A plot size of $6 \times 24$ m was used for each treatment with a 1 m buffer zone to separate treatments from each other.

### 2.3. Crop Husbandry

Seeds were sown with residual soil moisture as the area under cultivation is rainfed. The seedbed was prepared according to the treatment protocol and sowing was done using a winter seed drill with rows spaced 25 cm apart during both study years. The seeding rates of oats, barley and vetch were 80, 80 and 40 kg $ha^{-1}$, respectively. Before sowing, the cereal and legume seeds were mixed well in a 70:30 ratio. Nitrogen (N) at 80 kg $ha^{-1}$ and phosphorous (P) at 40 kg $ha^{-1}$ were incorporated before sowing in the form of urea and DAP.

### 2.4. Observations Recorded

Fodder Yield and Related Measurements

Ten plants were randomly selected from each plot at 50% heading and their height from base to tip was measured using a measuring tape and averaged. Similarly, the number of tillers/branches were counted from ten randomly selected plants and averaged to calculate the number of tillers/branches per plant. Ten plants were harvested at 50% heading, separated into leaves and stems, and weighed to calculate leaf-to-stem ratio. Plants within an area of 1 $m^2$ were harvested at 50% heading, weighed and converted into tons per hectare (t $ha^{-1}$) for green fodder or forage yield.

### 2.5. Quality Parameters

2.5.1. Crude Protein

Crude protein (CP) content was determined by the Kjeldahl method. One gram of oven-dried sample was mixed with 30 mL of concentrated $H_2SO_4$ and 5 g of digestion

mixture (10 parts $K_2SO_4$ + 1 part $CuSO_4.5H_2O$ + 1 part selenium metal powder). The mixture was digested at 420 °C in a Kjeldahl flask. After digestion, the flask was cooled and 100 mL of distilled water was added. Ten milliliters of 2% boric acid was added along with two drops of methyl red indicator. The $NH_3$ released was titrated with 0.1N $H_2SO_4$ to a light pink color at endpoint to find out nitrogen (N) contents. The obtained N content was multiplied by 6.25 to calculate the crude protein content [13].

$$\text{Crude Protein Percent} = \text{N content} \times 6.25$$

### 2.5.2. Neutral Detergent Fiber (NDF)

A 2 g sample was dried, mixed with 100 mL neutral detergent solution containing 0.5 g sodium sulfite, and refluxed for 1 h at 100 °C. The resulting suspension was filtered, rinsed once with hot water and washed twice with warm water. A crucible was weighed and its weight was recorded as $W_1$. The filtrate was transferred to the crucible and heated to constant weight in an oven (1–3 h). The weight of crucible and contents was recorded as $W_2$ [14]. NDF was calculated as:

$$\text{NDF (\%)} = \left( \frac{W2 - W3}{W1} \right) \times 100$$

where
    W1 = weight of the sample (g),
    W2 = weight of crucible and residue after drying (g), and
    W3 = weight of crucible and residue after incineration (g)

### 2.5.3. Acid Detergent Fiber (ADF)

A 2 g sample was dried, mixed with 100 mL of acid detergent solution containing acetyl trimethylammonium bromide, heated to the boiling point, and refluxed for about thirty minutes. The suspension was filtered, rinsed once with hot water, then washed twice more with warm water. The filtrate was transferred into a weighed crucible ($W_1$) and heated to constant weight in an oven (1–2 h). The weight of crucible and contents was recorded as $W_2$ [14]. ADF was calculated as:

$$\text{ADF(\%)} = \left( \frac{W2 - W3}{W1} \right) \times 100$$

where
    W1 = weight of the sample (g),
    W2 = weight of crucible and residue after drying (g), and
    W3 = weight of crucible and residue after incineration (g)

### *2.6. Competitive Indices*

### 2.6.1. Land Equivalent Ratio (LER)

The total land equivalent ratio, $LER_T$, was calculated by combining the legume partial LER ($LER_L$) and cereal partial LER ($LER_C$) according to the following [15]:

$$LER_T = LER_L + LER_C = Y_{IL}/Y_{SL} + Y_{IC}/Y_{SC}$$

where $Y_{IL}$ and $Y_{IC}$ are yields of intercropped legume and cereal per unit area, respectively. $Y_{SL}$ and $Y_{SC}$ are yields of solo-cropped legume and cereal per unit area, respectively.

### 2.6.2. Competitive Ratio (CR)

Competitive ratio is a method to measure the competition that exists between various species. It provides additional information about the competitive capacity of the crops, and serves as an index over K and actual yield loss [16]. It shows the ratio of distinct $LER_S$

of two crop species, and takes into account the proportions of the crops. The CRs were obtained by using the following formulas:

$$\text{CR cereal} = (\text{LERC}/\text{LERL})(Z\,lc/Z\,cl)$$

$$\text{CR legume} = (\text{LERL}/\text{LERC})(Z\,cl/Z\,lc)$$

### 2.6.3. Actual Yield Loss (AYL)

The AYL index gives accurate information regarding competition between and within crop species and the behavior of each species in mixed cropping [17]. It is comparable to the gain or loss in yield from intercropping compared to individual crops.

The AYL was obtained by using the following equations:

$$\text{AYL} = \text{AYL cereal} + \text{AYL legume}$$

where

$$\text{AYL cereal} = ((Y_{cl}/X_{cl})/(Y_c/X_c)) - 1$$

and

$$\text{AYL legume} = ((Y\,lc/X_{lc})/(Y_l/X_l)) - 1$$

where $X_{cl}$ represent the sown proportion of intercrop legume with cereal and $X_{lc}$ represent the cereal with legume.

### 2.7. Correlation and Regression Analysis

Correlation analysis refers to the mutual relationship or connection between two or more variables. Regression analysis is a statistical process that estimates the relationships among variables, especially when the focus is on the relationship between a dependent variable and one or more independent variables.

### 2.8. Economic Analysis

Experimental data was analyzed for economic effects using the methodology described in CIMMYT [18].

### 2.9. Statistical Analysis

Data collected on all parameters were analyzed using Statistix software (version 8.1) and analysis of variance (ANOVA). Treatment means were compared using the least significant difference (LSD) test at 5% probability level [19]. Correlation and regression analyses were performed using SPSS-17.

## 3. Results

The two-year field investigation (Table 1) showed a significant difference between the conservation and conventional tillage systems in terms of plant height. All test crops and combinations performed better with the conventional tillage system than the conservation tillage system. The tallest forage crop plants were recorded with the oats + vetch mixture followed by the barley + vetch mixture during both years. Whereas, shorter plants were recorded in pure stands of vetch. In pure stands, oat plants produced maximum height followed by barley and vetch. There was a 5% increase in oat plant height compared to barley and a 17% increase relative to vetch. In combination treatments, an increase of 2% was recorded in oats + vetch over barley + vetch.

The results showed that there was a significant effect of tillage system on the number of tillers/branches. The maximum number of tillers/branches was obtained with conventional tillage. An increase of 14% in conventional tillage over conservation tillage was recorded. All crops and their mixtures performed better with conventional tillage than conservation tillage in terms of number of tillers/branches.

**Table 1.** Plant height, number of tillers/branches, leaf: stem, dry matter yield of cereal–legume forages grown in pure stands and in combination under conservation and conventional tillage systems.

| Treatments | 2013–14 | | | 2014–15 | | |
|---|---|---|---|---|---|---|
| | PT | CT | Mean (C) | PT | CT | Mean (C) |
| **Plant height (cm)** | | | | | | |
| Oats | 103.25 a | 98.25 ab | 100.75 C | 99.25 a | 93.75 a | 96.5 C |
| Barley | 97.75 a | 92.50 bc | 95.12 D | 96.25 a | 91.00 ab | 93.6 D |
| Vetch | 87.00 b | 84.75 c | 85.87 E | 83.75 b | 82.00 b | 82.9 E |
| Barley + Vetch | 105.75 a | 102.25 a | 104.0 B | 102.75 a | 98.75 a | 100.8 B |
| Oats + Vetch | 106.75 a | 103.25 a | 105.0 A | 104.75 a | 101.50 a | 103.1 A |
| Mean (T) | 100.1 A | 97.3 B | | 97.35 A | 93.4 B | |
| LSD ($p \leq 0.05$) | C = 2.8; T = 4.2; C × T = 5.4 | | | C = 3.0; T = 4.2; C × T = 5.6 | | |
| **Number of tillers/branches** | | | | | | |
| Oats | 9.3 a | 11.0 a | 10.1 A | 7.4 cd | 8.5 b | 7.9 B |
| Barley | 7.3 d | 8.5 bc | 7.9 C | 6.3 e | 7.3 d | 6.8 C |
| Vetch | 5.3 e | 5.8 e | 5.5 D | 5.0 f | 6.5 de | 5.8 D |
| Barley + Vetch | 7.3 d | 8.0 cd | 7.6 C | 7.0 d | 8.1 bc | 7.6 B |
| Oats + Vetch | 8.0 cd | 8.9 b | 8.4 B | 8.5 b | 9.8 a | 9.1 A |
| Mean (T) | 7.4 B | 8.4 A | | 6.8 B | 8.0 A | |
| LSD ($p \leq 0.05$) | C = 0.56; T = 0.27; C × T = 0.79 | | | C = 0.47; T = 0.70; C × T = 0.66 | | |
| **Leaf-to-stem ratio** | | | | | | |
| Oats | 0.11 ns | 0.13 | 0.12 B | 0.10 ns | 0.12 | 0.11 B |
| Barley | 0.09 | 0.10 | 0.09 D | 0.09 | 0.10 | 0.09 D |
| Vetch | 0.08 | 0.09 | 0.09 E | 0.08 | 0.09 | 0.09 D |
| Barley + Vetch | 0.09 | 0.12 | 0.11 C | 0.09 | 0.12 | 0.10 C |
| Oats + Vetch | 0.11 | 0.14 | 0.13 A | 0.11 | 0.14 | 0.13 A |
| Mean (T) | 0.09 B | 0.11 A | | 0.10 B | 0.11 A | |
| LSD ($p \leq 0.05$) | C = 6.05; T = 0.01; C × T = ns | | | C = 5.75; T = 0.01; C × T = ns | | |
| **Green forage yield (t ha$^{-1}$)** | | | | | | |
| Oats | 41.63 c | 43.50 ab | 42.56 B | 39.88 c | 41.31 ab | 40.59 B |
| Barley | 40.25 d | 41.75 c | 41.00 C | 38.52 d | 39.97 c | 39.25 C |
| Vetch | 27.25 f | 29.38 e | 28.31 D | 25.91 e | 26.85 e | 26.38 D |
| Barley + Vetch | 41.47 c | 42.83 b | 42.15 B | 40.60 bc | 41.48 ab | 41.04 AB |
| Oats + Vetch | 42.88 b | 43.88 a | 43.38 A | 41.10 ab | 41.86 a | 41.48 A |
| Mean (T) | 38.69 B | 40.27 A | | 37.20 B | 38.29 A | |
| LSD ($p \leq 0.05$) | C = 0.72; T = 0.41; C × T = 1.02 | | | C = 0.63; T = 0.75; C × T = 0.89 | | |

PT = conservation tillage; CT = conventional tillage; C = crop treatments; T = tillage systems.

In pure stands and mixtures of forage crops, the maximum number of tillers/branches was observed in oats + vetch combinations during the second year, followed by pure stands of oats. In contrast, the lowest number of branches was recorded in vetch during both years (Table 1). In pure stands, oats showed the maximum number of tillers followed by barley and vetch. There was an increase of 17 and 41% in oats over barley and vetch, respectively. In mixtures, oats + vetch produced 20% more tillers/branches than barley + vetch.

The leaf-to-stem ratio of forage crops in conservation and conventional tillage was significantly different (Table 1). Conventional tillage showed the best results, with a 22% increase in leaf-to-stem ratio under conventional tillage compared to conservation tillage. In all forage crops, it was observed that the maximum leaf-to-stem ratio was recorded with the oats + vetch mixture followed by pure stands of oats. In contrast, the lowest leaf-to-stem ratio was observed in pure stands of vetch. Pure stands of oats had the highest leaf-to-stem ratio (33%), followed by barley and vetch. In cereal–legume combinations, oats + vetch produced the maximum leaf-to-stem ratio and there was an increase of 19% in the oats + vetch mixture compared to pure crop stands.

The results of the two-year investigation (Table 1) showed that statistically, the maximum green fodder yield in pure stands and mixtures was recorded with conventional

tillage compared to conservation tillage. Conventional tillage produced a 4% increase relative to the conservation tillage system. Among forage crops, the highest green fodder yield was recorded with the combination of oats and vetch, and the lowest was from pure stands of vetch during both years of the study. In pure stands, oats produced the highest yield of green fodder followed by barley and vetch; while in mixtures, oats + vetch produced a higher yield of green forage than barley + vetch.

There was also a significant difference between conservation and conventional tillage systems in crude protein (CP) content (Table 2). The highest CP was recorded with conservation tillage compared to conventional tillage. In forage crops, the maximum CP content was obtained in vetch, followed by barley + vetch mixture, pure stands of barley, and oats + vetch mixture. The lowest CP was in pure stands of oats. Vetch had the highest CP content in pure stands, followed by barley and oats. In mixtures, barley + vetch had a higher CP than oats + vetch.

**Table 2.** Crude protein, acid detergent fiber and neutral detergent fiber of cereal–legume forages grown as pure stands and as cereal–legume combinations under conservation and conventional tillage systems.

| Treatments | 2013–14 | | | 2014–15 | | |
|---|---|---|---|---|---|---|
| | **PT** | **CT** | **Mean (C)** | **PT** | **CT** | **Mean (C)** |
| **Crude Protein (g kg$^{-1}$)** | | | | | | |
| Oats | 87.00 b | 84.75 c | 85.9 E | 83.75 b | 82.00 b | 82.9 E |
| Barley | 103.25 a | 98.25 ab | 100.8 C | 99.25 a | 93.75 a | 96.5 C |
| Vetch | 106.75 a | 103.25 a | 105 A | 104.75 a | 101.50 a | 103.1 A |
| Barley + Vetch | 105.75 a | 102.25 a | 104 B | 102.75 a | 98.75 a | 100.8 B |
| Oats + Vetch | 97.75 a | 92.50 bc | 92.6 D | 96.25 a | 91.00 ab | 93.7 D |
| Mean (T) | 100 A | 96.2 B | | 97.35 A | 93.4 B | |
| LSD ($p \leq 0.05$) | C = 0.77; T = 1.2; C × T = 1.52 | | | C = 0.42; T = 0.18; C × T = 0.55 | | |
| **Acid detergent fiber (%)** | | | | | | |
| Oats | 28.29 a | 28.12 ab | 28.21 A | 26.49 a | 26.07 a | 26.28 A |
| Barley | 27.83 b | 27.86 ab | 27.85 B | 26.03 a | 26.31 a | 26.17 A |
| Vetch | 19.14 d | 19.11 d | 19.12 E | 18.09 c | 18.06 c | 18.07 D |
| Barley + Vetch | 20.99 c | 20.98 c | 20.98 D | 18.95 b | 18.93 b | 18.94 C |
| Oats + Vetch | 16.09 e | 16.29 e | 16.19 C | 15.04 d | 15.23 d | 15.14 B |
| Mean (T) | 22.47 | 22.47 | | 20.92 | 20.92 | |
| LSD ($p \leq 0.05$) | C = 0.31; T = ns; C × T = 0.55 | | | C = 0.51; T = ns; C × T = 0.68 | | |
| **Neutral detergent fiber (%)** | | | | | | |
| Oats | 42.44 a | 42.43 ab | 42.44 A | 41.52 a | 41.51 a | 41.51 A |
| Barley | 41.99 b | 42.05 ab | 42.02 B | 41.35 a | 41.31 a | 41.33 A |
| Vetch | 27.05 d | 27.19 d | 27.12 D | 26.13 c | 26.26 c | 26.19 C |
| Barley + Vetch | 37.97 c | 37.93 c | 37.95 C | 37.05 b | 36.98 b | 37.02 B |
| Oats + Vetch | 37.72 c | 37.62 c | 37.67 C | 36.85 b | 36.73 b | 36.79 B |
| Mean (T) | 37.43 | 37.44 | | 36.58 | 36.56 | |
| LSD ($p \leq 0.05$) | C = 0.29; T = ns; C × T = 0.41 | | | C = 0.23; T = ns; C × T = 0.33 | | |

PT = conservation tillage; CT = conventional tillage; C = crop treatments; T = tillage systems.

Table 2 shows that there was no significant effect of conservation and conventional tillage systems on acid detergent fiber (ADF) content. Among the forage crops, the highest %ADF was in oats grown as pure stands followed by pure stands of barley, oats + vetch and barley + vetch, while the lowest ADF was in pure stands of vetch. In pure stands, the maximum ADF was found in oats followed by barley and vetch, while in mixtures, oats + vetch had a higher ADF than barley + vetch. For neutral detergent fiber (NDF), there was no significant difference between conservation and conventional tillage systems (Table 2). The highest NDF was found in pure stands of oats followed by pure stands of barley, barley + vetch, oat + vetch, and the lowest was found in pure stands of vetch

(Table 2). In pure stands, the maximum NDF was found in oats followed by barley and vetch, while in mixtures, barley + vetch had a higher NDF than oats + vetch.

*Competitive Indices*

Data regarding the land equivalent ratio (LER) are presented in Table 3. For oats + vetch, the value of LER was 70:30, and it was the same for the barley + vetch combination. The total LER value was higher than 1.0, which shows the advantage of mixed cropping in producing a higher yield. We also found that conventional tillage performed better than conservation tillage in terms of LER. The barley + vetch combination performed better with conventional tillage than conservation tillage, followed by oats + vetch. The oats + vetch mixture also performed better under conventional tillage. Barley + vetch resulted in a 52% higher LER than a pure stand of barley. Similarly, oats + vetch resulted in 42% higher LER than a solo oat crop, which shows the advantage of intercropping over solo cropping.

**Table 3.** Land equivalent ratios of cereal–legume mixtures under conventional vs. conservation tillage systems during 2013–14 and 2014–15.

| Crops | Seeding Ratio | Conventional Tillage | Conservation Tillage | Mean |
|---|---|---|---|---|
| Oats + Vetch | 70:30 | 1.55 | 1.29 | 1.42 |
| Barley + Vetch | 70:30 | 1.58 | 1.47 | 1.52 |
| Mean | | 1.56 | 1.58 | |

The competitive ratio (CR) is an additional approach to evaluate competition among different crops. In oats + vetch intercropping, partial CR values were higher for oats under conventional and conservation tillage during both study years. The highest partial CR values of 4.37 and 5.31 were recorded for the oat–vetch combination under conventional and conservation tillage system, respectively (Figure 2). Similarly, in barley–vetch intercropping, partial CR values were higher in barley under conventional and conservation tillage system during both years. The highest partial CR values of 4.25 and 5.83 were recorded for barley in barley + vetch combination under conventional and conservation tillage, respectively. The differences in CR values in vetch under conventional and conservation tillage systems was negligible, which indicated the same contribution was offered by the legume to oats and barley in the intercropping system.

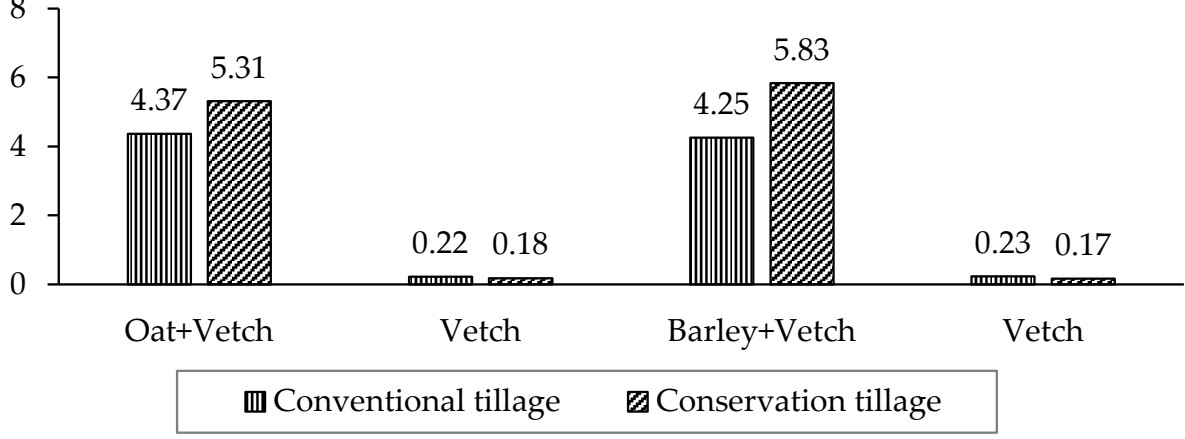

**Figure 2.** Competitive ratios recorded from oats + vetch and barley + vetch mixtures grown under conventional and conservation tillage systems.

Data regarding actual yield loss (AYL) showed that intercropped oats and barley were dominant over vetch under conventional and conservation tillage systems. AYL values of oat and barley were positive while in vetch it was negative, possibly due to the negative effects of oat and barley on intercropped legume. The yield declined to 77% and 83% in vetch when intercropped with oat under conventional and conservation tillage, respectively. It was 75% and 82% lower when intercropped with barley under conventional and conservation tillage respectively. Figure 3 shows that oats expressed 35% (1.35) and 9% (1.09) yield advantage in oat–vetch intercropping, compared to solo cropping under conventional and conservation tillage system, respectively. Barley expressed 39% (1.39) and 44% (1.44) yield advantage in barley–vetch intercropping system compared to solo cropping under conventional and conservation tillage, respectively.

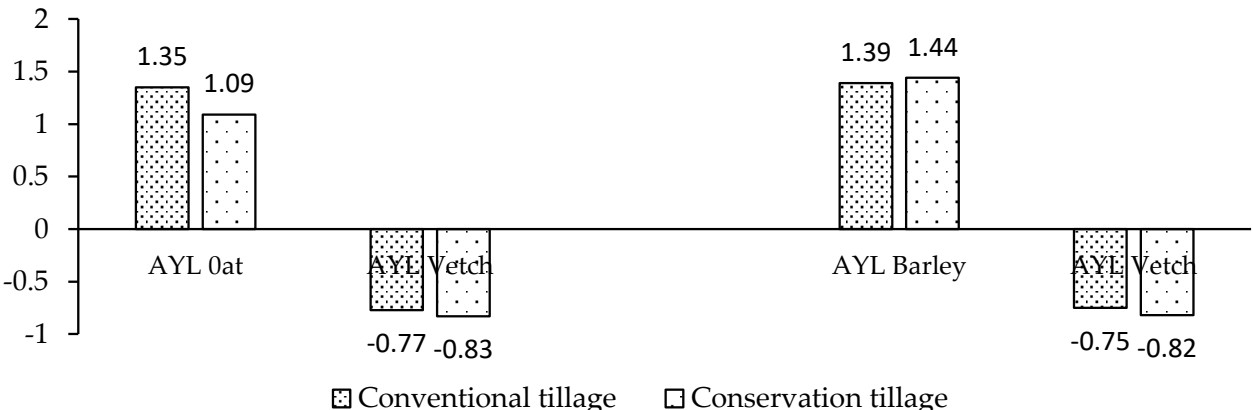

**Figure 3.** Actual yield loss recorded from oats + vetch and barley + vetch combinations grown under conventional and conservation tillage systems.

Correlation analysis among various variables including green forage yield, plant height, number of tillers/branches per plant, and leaf-to-stem ratio is presented in Table 4. Regression analysis showed that the numbers of tillers/branches per plant had a strong positive relationship ($R^2$ = 0.688) with green forage yield (Figure 4). The plant height showed a positive but weak relationship ($R^2$ = 0.298) with green forage yield (Figure 5). Economic analysis showed that the maximum benefit-to-cost ratio (BCR) was found under conventional tillage in oats + vetch mixture during both years of the study (Table 5).

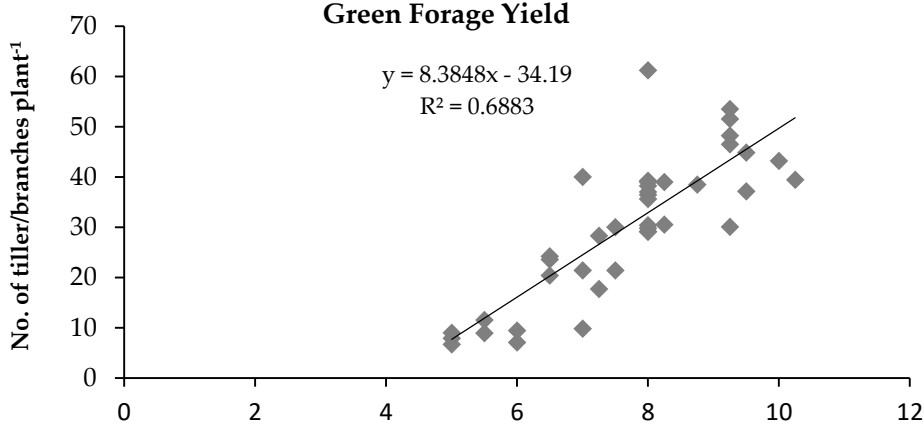

**Figure 4.** Relationship between average numbers of tillers/branches per plant and green forage yield during 2013–14 and 2014–15 under conservation and conventional tillage.

**Table 4.** Correlation among different variables such as green forage yield, plant height, number of tillers/branches per plant, and leaf-to-stem ratios during 2013–14 and 2014–15.

|  | Green Forage Yield | Plant Height | No. of Tillers/ Branches per Plant |
|---|---|---|---|
| Plant height | 0.5463 |  |  |
| No. of tillers/ branches per plant | 0.8296 | 0.6865 |  |
| Leaf-to-stem ratio | 0.5042 | 0.8203 | 0.6436 |

**Table 5.** Economic analysis of the effect of cereals and legumes grown in pure stands, as well as in combinations under conventional and conservation tillage systems.

|  | Treatments | Dry Matter Yield (t ha$^{-1}$) | Gross Income (Rs. ha$^{-1}$) | Fixed Cost (Rs. ha$^{-1}$) | Variable Cost (Rs. ha$^{-1}$) | Total Cost (Rs. ha$^{-1}$) | Net Benefits (Rs. ha$^{-1}$) | Benefit-Cost Ratio |
|---|---|---|---|---|---|---|---|---|
| | | | | **Year 1** | | | | |
| **PT** | Oats | 41.63 | 74,934 | 29,250 | 4198 | 33,448 | 41,486 | 2.24 |
| | Barley | 40.25 | 72,450 | 29,250 | 4198 | 33,448 | 39,002 | 2.17 |
| | Vetch | 27.25 | 49,050 | 29,250 | 4198 | 33,448 | 15,602 | 1.47 |
| | Barley + Vetch | 41.47 | 74,646 | 29,250 | 4198 | 33,448 | 41,198 | 2.23 |
| | Oats + Vetch | 42.88 | 77,184 | 29,250 | 4198 | 33,448 | 43,736 | 2.31 |
| **CT** | Oat | 43.5 | 78,300 | 29,250 | 1605 | 30,855 | 47,445 | 2.54 |
| | Barley | 41.75 | 75,150 | 29,250 | 1605 | 30,855 | 44,295 | 2.44 |
| | Vetch | 29.38 | 52,884 | 29,250 | 1605 | 30,855 | 22,029 | 1.71 |
| | Barley + Vetch | 42.83 | 77,094 | 29,250 | 1605 | 30,855 | 46,239 | 2.50 |
| | Oats + Vetch | 43.88 | 78,984 | 29,250 | 1605 | 30,855 | 48,129 | 2.56 |
| | | | | **Year 2** | | | | |
| **PT** | Oats | 39.88 | 71,784 | 29,250 | 4198 | 33,448 | 38,336 | 2.15 |
| | Barley | 38.52 | 69,336 | 29,250 | 4198 | 33,448 | 35,888 | 2.07 |
| | Vetch | 25.91 | 46,638 | 29,250 | 4198 | 33,448 | 13,190 | 1.39 |
| | Barley + Vetch | 40.6 | 73,080 | 29,250 | 4198 | 33,448 | 39,632 | 2.18 |
| | Oats + Vetch | 41.1 | 73,980 | 29,250 | 4198 | 33,448 | 40,532 | 2.21 |
| **CT** | Oats | 41.31 | 74,358 | 29,250 | 1605 | 30,855 | 43,503 | 2.41 |
| | Barley | 39.97 | 71,946 | 29,250 | 1605 | 30,855 | 41,091 | 2.33 |
| | Vetch | 26.85 | 48,330 | 29,250 | 1605 | 30,855 | 17,475 | 1.57 |
| | Barley + Vetch | 41.48 | 74,664 | 29,250 | 1605 | 30,855 | 43,809 | 2.42 |
| | Oats + Vetch | 41.86 | 75,348 | 29,250 | 1605 | 30,855 | 44,493 | 2.44 |

PT = conservation tillage; CT = conventional tillage.

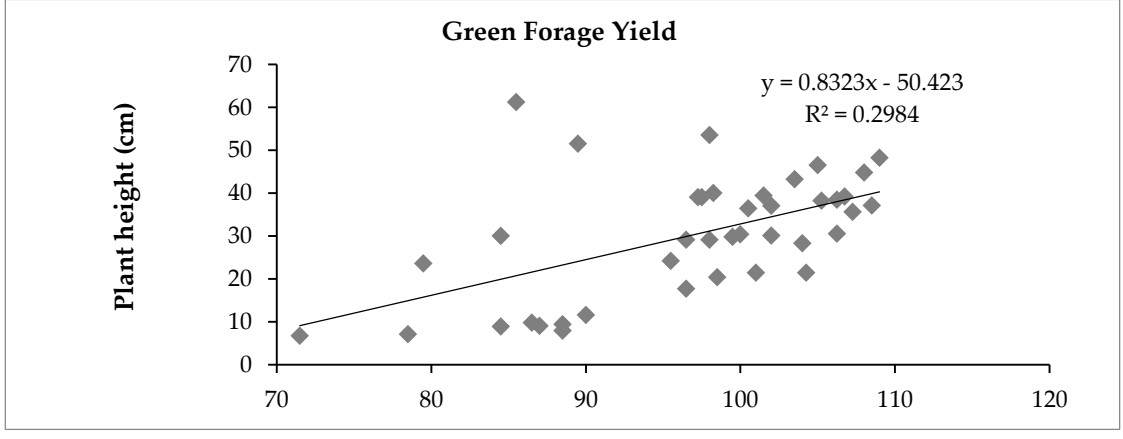

**Figure 5.** Relationship between average plant height and green forage yield during 2013–14 and 2014–15 under conservation and conventional tillage.

## 4. Discussion

In this study, response of winter cereal–legume forage and their mixtures under conventional and conservation tillage system was investigated and the most economical tillage practice for rainfed conditions was also identified. The results of study showed that tillage methods significantly influenced the number of tillers/branches per plant, leaf-to-stem ratio, green forage yield and dry matter yield, but there was no significant difference in plant height, crude protein contents, acid detergent fiber and neutral detergent fiber. However, conventional tillage system significantly influenced the number of tillers/branches per plant, leaf-to-stem ratio, green forage yield and dry matter yield, and performed better than conservation tillage system. In contrast, higher biomass and grain yield was observed in chisel-ploughed plots due to the loose and finer soil structure in deep tillage system due to the annual soil disturbance and pulverizing which improved the yield components and grain yield [20,21]. Wasaya et al. [21] also reported that tillage practices provide a favorable environment for plant growth and development and also for nutrients use due to lower soil bulk density and higher soil porosity under chisel-ploughed plots. Plant height was not significantly affected by conventional and conservation tillage systems. These results are in agreement with Keshavarzpour [22] who recorded non-significant effect of tillage treatments such as conservation and conventional systems on the plant height. No remarkable influence of conventional and conservation tillage systems was observed on CP, ADF and NDF. These results are in line with Baghdadi et al. [23] who reported that the effect of different tillage treatments including conventional and conservation on CP, NDF and ADF was not significant. In all forage crops and their mixtures, oat + vetch mixture performed better than other pure stands and mixtures in terms of plant height, number of tillers/branches per plant, leaf-to-stem ratio, green forage yield and dry matter yield. These results are in agreement with Ansar et al. [12] who reported the maximum plant height and leaf-to-stem ratio in oat–vetch mixture. Canan and Orak [24] observed the maximum numbers of tillers/branches, green fodder yield and dry matter yield in oat + vetch mixture. In case of CP, the maximum values were obtained in pure stand of vetch. These results are in line with Ansar et al. [12] who reported higher CP contents in pure stand of vetch than the other pure stands of crops, as well as their mixtures. The maximum values of ADF and NDF were obtained in pure stand of oat than other mixtures and legume vetch. These results are in line with Shoaib [25] who reported lower ADF in legumes as compared to sorghum, and Halil et al. [26] stated that legumes have less NDF value.

Total LER values were higher in both relay intercropping systems (Table 3), which exhibits the yield benefit of the intercropping system over sole cropping systems. Higher LER in intercropping compared with sole cropping might be due to the better utilization of land and environmental resources for crop growth and development under intercropping system [27]. These results are in agreement with Baink et al. [17] who reported that the LER was 80:20 in the case of pea–cereal mixtures and 60:40 in pea–triticale mixtures. The total LER value was higher than 1.00 which reflects a yield advantage of mix cropping over pure stands. The evaluation of loss or gain in yield due to other species could not be acquired through partial LER values. Partial AYL shows yield loss or gain by its sign and its value [16]. The economic analysis of conventional and conservation tillage system showed that a conventional tillage system provided more economical tillage than a conservation tillage system in arid conditions due to its more net benefit.

All crops performed better in the crop growth period of 2013–14 than 2014–15 in terms of plant height, numbers of tillers/branches per plant, leaf-to-stem ratio, green forage yield and dry matter yield. This could be the result of high rainfall (564.61 mm) received in early months of experimentation (October, November, December and January) which certainly provided a better environment for germination, growth and plant establishment in 2013–14 than the crop season of 2014–15 where only 178 mm rainfall was recorded in the same crop growth period. This clearly indicated that rainfall was not sufficient which resulted in less germination, growth and plant establishment.

## 5. Conclusions

The results of this two-year field study revealed that conventional tillage system performed better in terms of numbers of tillers/branches, leaf-to-stem ratio and green fodder yield than the conservation tillage system. The conventional and conservation tillage systems did not show significant difference from one another in case of fodder quality. In cereal–legume pure stand and mixtures, oat–vetch mixture performed better in terms of green fodder yield. Additionally, cereal–legume mixture improved fodder quality and maximum crude protein contents were observed in oat–vetch mixture. Cereal–legume mixture also improved land equivalent ratio and showed advantage of intercropping over sole cropping. Therefore, oat–vetch mixture and conventional tillage system might be the best option for rainfed farmers for improving fodder yield and quality.

**Author Contributions:** Conceptualization and methodology: S.S. and M.A.; software and validation: A.W., M.A., and S.S.; investigation and resources: S.S., and M.A., writing—original draft preparation: M.A., A.W., T.A.Y., and S.S.; reviewing and editing, W.S., M.A.R., M.S.I., M.S., M.B. and A.E.S. funding acquisition, W.S., A.E.S., M.S., M.B., A.M.E.-S. All authors have read and agreed to the published version of the manuscript.

**Funding:** The current work was funded by Taif University Researchers Supporting Project number (TURSP-2020/75), Taif University, Taif, Saudi Arabia.

**Acknowledgments:** All the authors are grateful to PMAS-Arid Agriculture University, Rawalpindi, Pakistan for providing funds to complete this research as a part of M. Phil thesis. The authors extend their appreciation to Taif University for funding current work by Taif University Researchers Supporting Project number (TURSP-2020/75), Taif University, Taif, Saudi Arabia. We thank Gary Bentley for his valuable help with English editing.

**Conflicts of Interest:** All the authors declare that they have no conflict of interest.

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
