# Peer review of "Influence of Tillage Systems and Cereals–Legume Mixture on Fodder Yield, Quality and Net Returns under Rainfed Conditions"

_sustainability, doi:10.3390/su13042172_

Round 1
Reviewer 1 Report
The manuscript titled “Influence of Tillage Systems and Cereals-Legume Mixture on Fodder Yield, Quality and Net Returns under Rainfed Condition” is fairly interesting and demanding and meets the objectives of the study. It highlights the importance of nutritional quality of fodder and economics. The author studied the impact of tillage systems on sole cereal crops as well as on combination of cereals and legume which represent a clear picture of the production and quality of different fodder mixture. The basic analyses required to judge the quality parameters are described sufficiently. The statistical procedures are well composed. There are some discrepancies in the manuscript which need to be addressed before publication.
Abstract:
Lines 20-22: Split the phrase into two sentences.
Livestock development…….fodder and supply. The mono-cropping system…deficient in protein.
Line 33: was observed……….were observed
Line 38: is recommended….are recommended
Introduction
Line 46-47: Not clear, rearrange the sentence
Line 47-48: rural community fulfill their dietary (meat, milk etc) as well as byproducts (leather, wool products) from livestock.
Line 56: a hardpan are observed…………hardpans are observed
Line 56: this hardpan…….These hardpans
Line 63: The farmers fed……The farmers feed
Line 75: rainfed region…rainfed regions
Materials and Methods:
Line 82: and the climatic data………..and two year climatic data
Line 115: Add reference for Kjeldahl method
Line 126: H2SO4….use subscript option for numeric
Line 136: it weight….its weight
Line 183: Combine both sentences
Statistical Analysis:
Line 189-91: An appropriate computer software…..delete it
Rewrite this section. Mentioning the purpose for using ANOVA and LSD. Also add software name on which ANOVA, LSD, Correlation and Regression analysis were performed.
Results & Discussion
Use taller plants or shorter plants instead of taller plant or shorter plant.
Avoid repetition of “All crops and their mixtures performed better in conventional tillage system than conservation tillage system”.
Results section is very lengthy. Rewrite it by avoid repetition and be more specific.
Discussion section is well-written
Author Response
Reviwer-1 comments:
Date: 28 January 2021
To Editor: Sustainability
Ref: Manuscript ID: sustainability- 1098025
Subject: Submission of a revised version of the manuscript: “Influence of Tillage Systems and Cereals-Legume Mixture on Fodder Yield, Quality and Net Returns under Rainfed Condition", for evaluation and possible publication in journal ‘Sustainability’
Thank you so much for sending the reviewers’ comments for the manuscript, which have allowed us for considerable improvement of the manuscript before publishing in ‘Sustainability’. We are happy to inform you that we have been able to address all of the reviewers’ comments. For clarity, the original comments and suggestions made the reviewers will appear in black colored text, while our response will appear in blue text. Please kindly note that all edits are shown in track changes mode in the manuscript. If unintentionally we have overlooked any issue, kindly let us know and we will rectify that.
We also confirm that all authors of the manuscript have read and approved the submission of the revised version to ‘Sustainability’.
Please find below our responses to reviewers’ comments:
The manuscript titled “Influence of Tillage Systems and Cereals-Legume Mixture on Fodder Yield, Quality and Net Returns under Rainfed Condition” is fairly interesting and demanding and meets the objectives of the study. It highlights the importance of nutritional quality of fodder and economics. The author studied the impact of tillage systems on sole cereal crops as well as on combination of cereals and legume, which represent a clear picture of the production and quality of different fodder mixture. The basic analyses required to judge the quality parameters are described sufficiently. The statistical procedures are well composed. There are some discrepancies in the manuscript, which need to be addressed before publication.
Response: Thanks for appreciation
Comment 1: Abstract: Lines 20-22: Split the phrase into two sentences. Livestock development…….fodder and supply. The mono-cropping system…deficient in protein.
Response: lines 20-22 has been corrected.
Line 33: was observed……….were observed
Response: Correction has been made
Line 38: is recommended….are recommended
Response: Corrected
Introduction
Line 46-47: Not clear, rearrange the sentence
Response: Line 46-47 has been rearranged
Line 47-48: rural community fulfill their dietary (meat, milk etc) as well as byproducts (leather, wool products) from livestock.
Response: line 47-48 has been rephrased
Line 56: a hardpan are observed…………hardpans are observed
Response: it has been corrected.
Line 56: this hardpan…….These hardpans
Response: Corrected
Line 63: The farmers fed……The farmers feed
Response: Previous sentence is good
Line 75: rainfed region…rainfed regions
Response: Corrected
Materials and Methods:
Line 82: and the climatic data………..and two year climatic data
Response: Added
Line 126: H2SO4….use subscript option for numeric
Response: This is already in subscript form
Line 136: it weight….its weight
Response: ok
Line 183: Combine both sentences
Response: Both sentences has been combined
Statistical Analysis:
Line 189-91: An appropriate computer software…..delete it. Rewrite this section. Mentioning the purpose for using ANOVA and LSD. Also add software name on which ANOVA, LSD, Correlation and Regression analysis were performed.
Response: Rephrased as suggested
Results & Discussion
Use taller plants or shorter plants instead of taller plant or shorter plant.
Response: sentence has been modified.
Avoid repetition of “All crops and their mixtures performed better in conventional tillage system than conservation tillage system”.
Response: Sentence has been rephrased
Results section is very lengthy. Rewrite it by avoid repetition and be more specific.
Response: Results section has been rewritten
Discussion section is well-written
Response: Thanks for appreciation

Reviewer 2 Report
I have any remarks.
The work aimed at to study the effect of tillage systems and cereals-legume mixture on fodder yield, quality and net returns under rained condition. The work deals with very actual problem in the agriculture. The paper is accurate in every chapter/abstract, Introduction, Material and methods ect./. The findings after two year experimental work are of benefit for science and farmers also.
Author Response
Reviwer-2 comments:
Date: 28 January 2021
To Editor: Sustainability
Ref: Manuscript ID: sustainability- 1098025
Subject: Submission of a revised version of the manuscript: “Influence of Tillage Systems and Cereals-Legume Mixture on Fodder Yield, Quality and Net Returns under Rainfed Condition", for evaluation and possible publication in journal ‘Sustainability’
Thank you so much for sending the reviewers’ comments for the manuscript, which have allowed us for considerable improvement of the manuscript before publishing in ‘Sustainability’. We are happy to inform you that we have been able to address all of the reviewers’ comments. For clarity, the original comments and suggestions made the reviewers will appear in black colored text, while our response will appear in blue text. Please kindly note that all edits are shown in track changes mode in the manuscript. If unintentionally we have overlooked any issue, kindly let us know and we will rectify that.
We also confirm that all authors of the manuscript have read and approved the submission of the revised version to ‘Sustainability’.
Please find below our responses to reviewers’ comments:
The work aimed at to study the effect of tillage systems and cereals-legume mixture on fodder yield, quality and net returns under rained condition. The work deals with very actual problem in the agriculture. The paper is accurate in every chapter/abstract, Introduction, Material and methods ect./. The findings after two-year experimental work are of benefit for science and farmers also.
Dear reviewer
We would like to acknowledge your contribution explicitly. Thank you very much for your kind comments on the manuscript. Those comments are very helpful for improving the manuscript. We have considered the comments and made correction. The corrections were in track in the revised version.
